# Functional Identification of *HhUGT74AG11*—A Key Glycosyltransferase Involved in Biosynthesis of Oleanane-Type Saponins in *Hedera helix*

**DOI:** 10.3390/ijms25074067

**Published:** 2024-04-05

**Authors:** Han Yu, Jun Zhou, Jing Zhang, Xinyi He, Siqing Peng, Hao Ling, Zhuang Dong, Xiangyang Lu, Yun Tian, Guiping Guan, Qi Tang, Xiaohong Zhong, Yuedong He

**Affiliations:** 1College of Horticulture, Hunan Agricultural University, Changsha 410128, China; yhan2023@stu.hunau.edu.cn (H.Y.); zhou_jun2023@stu.hunau.edu.cn (J.Z.); jingzhang0711@stu.hunau.edu.cn (J.Z.); hexinyi@stu.hunau.edu.cn (X.H.); pengsiqing@stu.hunau.edu.cn (S.P.); linghao1996@stu.hunau.edu.cn (H.L.); dongzhuang@stu.hunau.edu.cn (Z.D.); xh_zhong@hunau.edu.cn (X.Z.); 2College of Bioscience and Biotechnology, Hunan Agricultural University, Changsha 410128, China; xiangyanglu@hunau.edu.cn (X.L.); tianyun@hunau.edu.cn (Y.T.); guanguiping@hunau.edu.cn (G.G.)

**Keywords:** UDP-glycosyltransferase, substrate hybridity, biosynthesis, oleanane-type saponin, *Hedera helix*

## Abstract

*Hedera helix* is a traditional medicinal plant. Its primary active ingredients are oleanane-type saponins, which have extensive pharmacological effects such as gastric mucosal protection, autophagy regulation actions, and antiviral properties. However, the glycosylation-modifying enzymes responsible for catalyzing oleanane-type saponin biosynthesis remain unidentified. Through transcriptome, cluster analysis, and PSPG structural domain, this study preliminarily screened four candidate UDP-glycosyltransferases (UGTs), including Unigene26859, Unigene31717, CL11391.Contig2, and CL144.Contig9. In in vitro enzymatic reactions, it has been observed that Unigene26859 (*HhUGT74AG11*) has the ability to facilitate the conversion of oleanolic acid, resulting in the production of oleanolic acid 28-*O*-glucopyranosyl ester. Moreover, *HhUGT74AG11* exhibits extensive substrate hybridity and specific stereoselectivity and can transfer glycosyl donors to the *C*-28 site of various oleanane-type triterpenoids (hederagenin and calenduloside E) and the *C*-7 site of flavonoids (tectorigenin). Cluster analysis found that *HhUGT74AG11* is clustered together with functionally identified genes *AeUGT74AG6*, *CaUGT74AG2*, and *PgUGT74AE2*, further verifying the possible reason for *HhUGT74AG11* catalyzing substrate generalization. In this study, a novel glycosyltransferase, *HhUGT74AG11*, was characterized that plays a role in oleanane-type saponins biosynthesis in *H. helix,* providing a theoretical basis for the production of rare and valuable triterpenoid saponins.

## 1. Introduction

*Hedera helix* is a perennial evergreen vine belonging to Araliaceae, *Hedera* [1]. It is widely cultivated around the world. As a traditional medicinal plant, it was included in the European Pharmacopoeia in 2010 [2]. Modern pharmacology indicates that oleanane-type saponins are the main active ingredient of *H. helix*, such as hederacoside-C, oleanolic acid, and hederagenin [3]. Oleanane-type saponins have extensive pharmacological effects, especially gastric mucosal protection [4], autophagy regulation actions [5], and antiviral properties [6]. Oleanane-type saponins are widely applied in clinical practice to treat various respiratory system diseases [7]. In recent years, as patients with respiratory system diseases such as COVID-2019 and influenza are soaring, scholars predict that the demand for *H. helix* extract will also increase sharply year by year [8]. However, due to the limitations of extracting oleanane-type saponins from plants and the difficulty of chemical synthesis, biosynthesis is expected to be an alternative with broadly applicable prospects for the production of oleanane-type saponins.

Oleanane-type saponins biosynthesis pathway begins at the common stage of terpene precursors and ultimately produces isopentenyl pyrophosphate (IPP) through the cytoplasmic mevalonic acid (MVA) pathway and the plastid Mevalonate (MEP) pathway [9]. Diphosphate synthase (FPS) converts dimethylallyl diphosphate (DMAPP) and IPP into farnesyl diphosphate (FPP). Squalene synthase (SS) [10] and squalene epoxidase (SE) catalyze the reaction between the two FPP molecules to form 2,3-oxidosqualene. The second stage is the formation of triterpenoid skeletons. The cyclization of 2,3-oxidosqualene by oxidosqualene cyclases (OSCs) produces various tetracyclic and pentacyclic triterpenoid scaffolds [11]. The third stage includes oxidative modification based on cytochrome P450 (CYP450) and glycosylation modification of UDP-glycosyltransferases (UGTs) with uridine diphosphate glucose (UDP-Glc) as the donor [12]. Ultimately, oleanane-type saponins with diverse structures are generated.

The UGT enzyme has a vital function in the production of oleanane-type saponins, enhancing their chemical stability and solubility in water while decreasing their toxicity and activity [13]. Genes of the UGT74 subfamily play a crucial role in this process. Previous studies have shown that UGT74 subfamily members play a significant role in the glycosylation of terpenoid saponins in Araliaceae plants. For example, *Panax ginseng PgUGT74AE2* can glycosylate protopanaxadiol-type ginsenosides at the *C*3-OH site [14]; *Aralia elata AeUGT74AG6* can conduct glucosylation at the *C*-28 site of various triterpenoid precursors and generate multiple triterpenoid saponin compounds [15]. Nonetheless, the specific catalytic function in the production of oleanane-type saponins in *H. helix* remains uncertain for any member within the UGT74 subfamily.

In this study, four candidate UGT genes potentially involved in the biosynthesis of oleananes-type triterpenoids in *H. helix* were preliminarily screened through transcriptome, cluster analysis, and the PSPG domain. Using the *Escherichia coli* expression system and protein purification methods, the in vitro enzymatic activity of the candidate UGT genes was tested through precursor feeding assays. The results showed that only Unigene26859 (*HhUGT74AG11*) was active toward oleanolic acid. In addition, considering the diversity of the structures of oleananes-type triterpenoids, we also tested a variety of triterpenoid compounds and flavonoids, and *HhUGT74AG11* exhibited broad substrate promiscuity. Combining cluster analysis, we further discussed the reasons for the interesting phenomenon of the functional evolution of *HhUGT74AG11* and the structural diversification of secondary metabolites. In summary, this study cloned and functionally identified the glycosyltransferase *HhUGT74AG11* from *H. helix*, revealing its outstanding contribution to the biosynthesis, laying the foundation for the development of drugs with diverse structures of oleananes-type triterpenoids.

## 2. Results

### 2.1. UGT Candidate Gene Screening

In order to screen for key UGTs, this study first analyzed and screened 82 relatively highly expressed unigenes in the transcriptome data of *H. helix*, which were related to terpenoid metabolism and synthesis (Figure 1a). A hierarchical clustering of expression profiles of all unigenes was performed using standardized FPKM (Figure 1b and Appendix A). Significant differences were observed between tissues. Notably, with the exception of HMBPPS, MECDPS, and HDR, the stability of gene expressions related to the MEP and MVA pathways is higher in the roots. Therefore, in order to further narrow down the range of candidate UGTs, this study focused on Unigene26859, Unigene31717, CL11391.Contig2, and CL144.Contig9, which were relatively highly expressed in the roots.

Protein sequences from *H. helix* candidate UGTs and *Arabidopsis thaliana UGT74* (https://www.ncbi.nlm.nih.gov/search/all/?term=Arabidopsis+thaliana+UGT74 accessed on 15 January 2024) were used to build a phylogenetic tree (Figure 2a and Appendix A). It can be seen that Unigene26859 is highly homologous with five *AtUGT74* genes. Using the MEME tool, it is further confirmed that the candidate UGT protein sequences all contain the characteristic PSPG Box (Figure 2b), which is highly consistent with the PSPG box of four *AtUGT74* genes.

### 2.2. In Vitro Enzyme Activity Determination of UGTs

SDS-PAGE was employed to detect the expression and purification effect of candidate UGT proteins. All UGT proteins were induced for expression. The target band was single, and the molecular weight was consistent with the predicted result (Appendix A). The in vitro catalytic activity of candidate UGTs was detected by feeding the precursor oleanolic acid and glycosyl donor UDP-Glc (Figure 3a). The findings indicate that only Unigene26859 exhibits glycosylation with oleanolic acid. Furthermore, the retention time and fragment ions of the substance align with the established oleanolic acid 28-*O*-glucopyranosyl ester (RT = 7.03 min [(-)-mode], *m*/*z* = 617.52) (Figure 3b). Unigene26859 was named *UGT74AG11* (accession number: OR902350) by the UGT Naming Committee. The above results indicate that *HhUGT74AG11* is the UGT identified from *H. helix* and has the ability to perform glycosylation at the *C*-28 site.

### 2.3. Substrate Hybridity of HhUGT74AG11

In order to probe the substrate hybridity of *HhUGT74AG11*, using UDP-Glc as the sugar donor, this study carried out various substrate activity tests with different structures on purified *HhUGT74AG11* (Figure 4). Triterpenoid test substrates contain **1** oleanolic acid, **2** hederagenin, **3** echinocystic acid, and **4** calenduloside E. Flavonoids contain **5** quercetin, **6** luteolin, **7** baicalein, **8** phloretin, **9** genistein, and **10** tectorigenin. The experiment results demonstrate that the *HhUGT74AG11* enzyme exhibits strong hybridity and high catalytic activity toward tectorigenin, oleanolic acid, and hederagenin. In addition, when fed with **4** calenduloside E, the product **4a** chikusetsusaponin IVa (RT = 4.68 min [(-)-mode], *m*/*z* = 793.63) with multiple glycosides was still observed, implying that *HhUGT74AG11* can catalyze both aglycones and glycosides. All reactions of test substrates were analyzed using UPLC-ESI-MS (Appendix A).

### 2.4. Cluster Analysis of HhUGT74AG11

To investigate systematic clustering relationships between *HhUGT74AG11* and other plant UGT gene families, several UGTs from different species were selected and phylogenetically related (Figure 5 and Appendix A). Based on sequence homology, UGTs are divided into 14 branches (A-N). Along the evolutionary path of higher plants, branches A, D, E, and L have experienced greater expansion compared with other branches. As expected, *HhUGT74AG11* is classified as the L branch, which has high homology to *AeUGT74AG6* and *CaUGT74AG2* and is closely related to UGTs that have been confirmed to have flavonoid catalytic activity.

## 3. Discussion

UGTs are crucial for plant growth, development, and environmental adaptation, as they regulate glucose metabolism, maintain homeostasis, and influence secondary metabolites [13]. By investigating important UGT genes related to terpenoid production in *Arabidopsis thaliana* [16], *Nicotiana tabacum* [17], *Glycine max* [18], *Panax ginseng* [19], and *Panax notoginseng* [20], the possibility of utilizing synthetic biology to generate terpenes and glycosides of significant medicinal importance has progressively emerged. Research progress has been made in the heterologous production of terpenes and glycosides. Significant advancements have been made, particularly in the biosynthesis of natural sweeteners [21]. Glycosylation, as a critical modification step, has received widespread attention, providing new insights into the oleanane-type saponin biosynthesis.

This study initially analyzed the expression profiles of 82 unigenes associated with oleanane-type saponins biosynthesis using normalized FPKM values. Compared with leaves, unigenes involved in the common “upstream steps” of terpenes and the formation of triterpenoid parent nuclei show higher expression levels in roots (Figure 1). The same results have been observed in plants of the Araliaceae family, such as *Panax ginseng* squalene oxide cyclase (*PNY1*) [22] and *Panax notoginseng CYP72A938* [15], all of which exhibit high specificity of tissue expression. Therefore, it is speculated that the key UGT implicated in the biosynthetic process of *H. helix* has a similar gene expression pattern. Subsequently, by combining cluster analysis and PSPG domain analysis (Figure 2), this study narrowed down the screening range of key UGT members. Through biochemical experiments (Figure 3), the glycosyltransferase *HhUGT74AG11*, responsible for catalyzing oleanolic acid 28-*O*-glucopyranosyl ester, was finally identified in vitro. As expected, *HhUGT74AG11* is highly expressed in the roots of *H. helix* and displays unique catalytic activity in vitro.

*H. helix* is considered an excellent source of triterpenoid saponins and flavonoids [23]. Therefore, this study tested the substrate and product specificity of *HhUGT74AG11* through various oleanane-type triterpenoids and flavonoids (Figure 4 and Appendix A). Interestingly, *HhUGT74AG11* can catalyze various triterpenoid glycosides and triterpenoid saponins and presents significant activity toward a specific type of flavonoid. Members of the UGT74 subfamily are generally found in reports on triterpenoid saponin biosynthesis, such as *Saponaria vaccaria SvUGT74M1* [24], *Siraitia grosvenorii UGT74AC1* [25], and *Nigella sativa NsUGT74Q1* [26]. Some studies have published UGT74 subfamily members involved in flavonoid glycoside biosynthesis, such as *Oryza sativa OsUGT74* [27]. It is worth discussing that the strong hybridity of the UGT74 subfamily may be because ancestral enzymes on glycosylation functional branches help plants adapt to complex and changing chemical environments through the evolution and functional differentiation of gene families.

Plant glycosyltransferases belong to a large enzyme family with varied functions. Taking into account the variety of sequences and functions, phylogenetic trees are usually constructed to compare gene family members and determine similarities and differences [28]. Cluster analysis (Figure 5) indicates that, in the evolution of higher plants, branches A, D, E, and L appear to have experienced greater expansion compared with other branches, aligning with the more prominent branches observed in the evolutionary tree [29]. In branch A, multiple members have been identified as being implicated in the biosynthesis of triterpenoid saponins. For instance, *GmUGT91H4* can use rhamnose as a donor to convert soyasaponin III into soyasaponin I [30]. Some members of the UGT73 subfamily in the D branch and members of the UGT71 subfamily in the E branch can participate in triterpenoid saponin biosynthesis [31]. The G branch is considered to be beneficial for the biosynthesis of natural sweeteners, such as *Stevia rebaudiana SrUGT85C2*, which promotes the conversion of steviol to rebaudioside A [32]. *HhUGT74AG11* clusters with *AeUGT74AG6* [15], *PgUGT74AE2* [21], and *CaUGT74AG2* [33], and belongs to the L branch. Research on ginseng triterpenoid saponins has identified that *PgUGT74AE2* can glycosylate flavonoids, suggesting that *HhUGT74AG11* has a similar function. Meanwhile, this study observed that the two major branches responsible for triterpenoid and flavonoid glycosylation had a close phylogenetic relationship, and some members of the two branches were characterized by multiple functions. It is consistent with the strong substrate hybridity found in *HhUGT74AG11*. However, a more accurate and detailed differentiation and integrated function evolution still needs further investigation.

From a protein microscopic perspective, due to insufficient information on the crystal structure of UGT74 protein and its complexes, the fundamental sites determining the substrate specificity and hybridity of *HhUGT74AG11* are difficult to recognize. A recent study reported a special phenolic D-apiose transferase *GuApiGT* in licorice [34]. This study investigated the D-apiose donor selectivity mechanism of *GuApiGT* through crystal structure analysis, molecular docking and kinetic simulation, theoretical calculations, and mutation experiments. Therefore, the substrate hybridity and exhaustive catalytic mechanism of *HhUGT74AG11* still need verification through theoretical calculations and molecular modifications after obtaining its chemical crystal and precise 3D structure. Additionally, further research on the broader sugar donor selectivity of *HhUGT74AG11* is critical, which will provide assistance for a comprehensive understanding of regional selectivity and stereoselectivity in glycosylation reactions.

## 4. Materials and Methods

### 4.1. Botanical Substances and Compounds

The plant parts used in the experiment were obtained from a yearly ivy plant cultivated in the Traditional Chinese Medicine Resources and Development Department’s greenhouse at Hunan Agricultural University (HUNAU). The plant’s roots, stems, and leaves were harvested. The samples were rapidly cryopreserved using liquid nitrogen and then stored at −80 °C for later analysis. There were three biological replicates for each sample.

Various chemicals, such as echinocystic acid, oleanolic acid, calenduloside E, UDP-Glc, hederagenin, quercetin, luteolin, baicalein, phloretin, genistein, tectorigenin, oleanolic acid 28-*O*-glucopyranosyl ester, hederagenin 28-*O*-glucopyranosyl ester, echinocystic acid 28-*O*-glucopyranosyl ester, and chikusetsusaponin IVa were acquired from the Taoshu Biotechnology Company (Shanghai, China) or the Yuanye Biotechnology Company (Shanghai, China). All standard reagents had a purity of over 98%. Resin that had been purified and labeled as GST was purchased from Huiyan Biotechnology Co., Ltd. (Wuhan, China). Anhui Tiandi High Purity Solvent Co., Ltd. (TEDIA, Anhui, China) provided the acetonitrile, methanol, and formic acid used in high-performance liquid chromatography (HPLC) or liquid chromatography-electrospray ionization mass spectrometry (UPLC-ESI-MS) analyses. The OK Clon DNA Connection Kit was derived from Accurate Biotechnology Co., Ltd. (Changsha, China). Unless specified, other chemicals and reagents came from Sangon Biotech Co., Ltd. (Shanghai, China).

### 4.2. Examining the Transcriptome and Discovering Genes Associated with Triterpenoid Synthesis

The sequencing of the library preparations was conducted on an Illumina HiSeq 4000 platform, producing paired-end reads. To derive clean data, sequences with adapter contamination, poly-N stretches, and those of inferior quality were excised from the raw data. High-throughput sequencing and de novo assembly of *H. helix* transcriptome were based on previous research results [35]. The information was transferred to the SSR database, which is managed by the National Center for Biotechnology Information (NCBI) in the United States. The access number for this upload is SRR5369456. Further examination was conducted on all unigenes identified as associated with the triterpenoid biosynthesis pathway, utilizing databases including Nr, Nt, Swissprot, COG, GO, and KEGG. The FPKM method was used to normalize the expression level of every unigene. Candidate UGT full-length coding sequences were identified by clustering with *Arabidopsis thaliana* UGT (*AtUGT74*).

### 4.3. Bioinformatics Analysis

The ORF sequence of *HhUGT74AG11* was acquired through the utilization of the ORF Finder program (http://www.ncbi.nlm.nih.gov/projects/gorf/ accessed on 15 January 2024). The ExPASy Bioinformatics Resource Portal Website (http://au.expasy.org/ accessed on 15 January 2024) was adopted to analyze the molecular weight of *HhUGT74AG11* protein, and MEME software (https://meme-suite.org accessed on 15 January 2024) was employed to analyze and predict the PSPG structural domain of *HhUGT74AG11* protein.

Multiple UGTs from different species were selected and compared with *HhUGT74AG11* for protein sequence alignment. All protein sequences came from the GenBank database of NCBI. To construct the phylogenetic tree, the Neighbor-Joining approach from MEGA X [36] was employed. The specific settings were as follows: the guidance value was set to 1000 repetitions, and the p-distance method was set to delete pairwise options to handle gaps between protein sequences. This multi-gene tree has been displayed using the online program iTOL (https://itol.embl.de/ accessed on 15 January 2024).

### 4.4. Total RNA Extraction, cDNA First-Strand Synthesis, and Gene Cloning

Total RNA was extracted from each sample using the Plant RNA Extraction Kit (Tiangen, Beijing, China). To prepare single-strand cDNA through reverse transcription, the cDNA synthesis kit (Takara, Dalian, China) was utilized. Using specific primers (Appendix A), the CDS sequence of potential UGTs was amplified with cDNA serving as a template. The PCR protocol included an initial elongation phase at 95 °C for 3 min, succeeded by 35 rounds of separation at 95 °C for 15 s, pairing at 58 °C for 30 s, and elongation at 72 °C for 1 min. The last stage of the extension was carried out at a temperature of 72 °C for a duration of 5 min.

### 4.5. Expression of Escherichia coli and Protein Separation and Purification

The candidate UGT target fragments were homologously recombined between the restriction sites of the pGEX-4T-1 vector, which are *BamH* I and *EcoR* I, using the OK Clon DNA ligation kit (Accurate biotechnology, Changsha, China). Afterward, they were introduced into *Escherichia coli* BL21 (DE3) cells and grown in Luria–Bertani (LB) liquid medium with the addition of ampicillin (100 μg/mL) at a temperature of 37 °C. The induction of recombinant protein expression occurred by incubating 0.5 mM isopropyl β-D-thiogalactoside at a temperature of 16 °C and a speed of 160 rpm for a duration of 20 h, during which the OD value of 600 decreased to a range of 0.6 to 0.8. After being spun at 5000 revolutions per min for 10 min at a temperature of 4 °C, the cells were collected and suspended in a balanced solution containing 140 mM of sodium chloride, 2.7 mM of potassium chloride, 10 mM of sodium dihydrogen phosphate, 1.8 mM of potassium dihydrogen phosphate, and a pH of 7.3. Following the ultrasonic disruption, the fusion solution was passed to a column with GST agarose affinity and subsequently subjected to centrifugation. Validation of the target protein labeled with GST was performed through SDS-PAGE after obtaining it using elution buffers consisting of 50 mM Tris-HCl (pH 8.0) and 10 mM glutathione (GSH).

### 4.6. In Vitro Enzyme Activity Determination

The assay to measure the enzyme activity of the recombinant *HhUGT74AG11* was conducted in a reaction mixture of 200 μL, which consisted of 50 mM Tris-HCl (pH 7.0), 0.1 mM MgCl_2_, 0.1 mM substrate for the receptor, 1 mM UDP-Glc, and 10 μg of the protein that had been purified. The incubation was carried out at 30 °C for 12 h. The reaction was stopped by adding 200 μL of 80% methanol. The negative control reaction used boiled recombinant proteins (100 °C and 15 min). After centrifugation at a speed of 12,000 revolutions per min for a duration of 2 min, the reaction mixture was then passed through a membrane filter with a pore size of 0.22 μm. Then, machine detection was carried out on it with HPLC or UPLC-ESI-MS.

### 4.7. UPLC-ESI-MS Analysis

The examination of flavonoid reaction products was carried out utilizing HPLC (Thermo Fisher, Waltham, MA, USA) equipped with a WondaCract ODS-2 C18 column (4.6 × 250 mm; particle size 5.0 μm granularity; Shimadzu, Japan). Solvent A, which was acetonitrile, and solvent B, which was a mixture of 0.1% formic acid in water, made up the mobile phase. The elution process is carried out in the following manner: solvent A is used at a concentration of 20% for 0 min, followed by 50% for 10 min, and finally 100% for 18 min. The rate of flow is adjusted to 1.0 milliliters per min, accompanied by an injection volume of 5 microliters, while maintaining a temperature of 35 °C. Spectral detection was performed at 269 nm using a diode array detector (DAD).

Reaction products were analyzed by UPLC-ESI-MS after storage of the Shim-pack GIST-HP C18 column at 35 °C. The separation of Acetonitrile (solvent A) and 0.1% formic acid in water (solvent B) occurred with a flow rate of 0.2 mL/min, while maintaining an injection volume of 1 μL. The experimental conditions were as follows: 0 min with 5% solvent A, 0.5 min with 5% solvent A, 4.5 min with 95% solvent A, 7.5 min with 95% solvent A, 7.6 min with 5% solvent A, and 10.5 min with 5% solvent A. All mass spectrometry data were acquired using the TSQ quantification ™ Triple Quadrupole Mass Spectrometer fitted with an ESI probe inlet (Thermo Fisher, Waltham, MA, USA). The parameters for the ESI source are as follows: triterpenoids are examined using negative ionization mode; flavonoids are examined using positive ionization mode; the spray voltage is set at 3.5 kV; collision energy is set at 35 V; and the scanning range is from 260 to 850 *m*/*z*. The flow of the auxiliary gas (also known as sheath gas) was adjusted to 40 arb, while the flow of the auxiliary/scavenging gas (specifically nitrogen) was set to 10 arb. The temperature of the auxiliary gas heater was kept constant at 300 °C, and the capillary temperature was set to 320 °C.

## 5. Conclusions

In summary, this study combined expression patterns, cluster analysis, and biochemical experiments of genes related to triterpenoid synthesis to screen and identify a glycosyltransferase implicated in the biosynthetic process of oleanane-type saponins of *H. helix*, *HhUGT74AG11*. In vitro enzymatic analysis discloses that *HhUGT74AG11* can catalyze oleanolic acid to produce 28-*O*-glucopyranosyl ester and catalyze various triterpenoids and flavonoids, including hederagenin, calenduloside E, and tetragenin. These findings verify that *HhUGT74AG11* has broad substrate heterozygosity and specific stereoselectivity. In addition, the possible reasons for the multifunctionality of *HhUGT74AG11* were further explored from the perspective of gene structure and functional clustering. This study probed into the primary glycosyltransferase implicated in the biosynthesis of various oleanane-type saponins, laying a foundation for synthetic biology research on structurally diverse triterpenoid saponins.

## Figures and Tables

**Figure 1 ijms-25-04067-f001:**
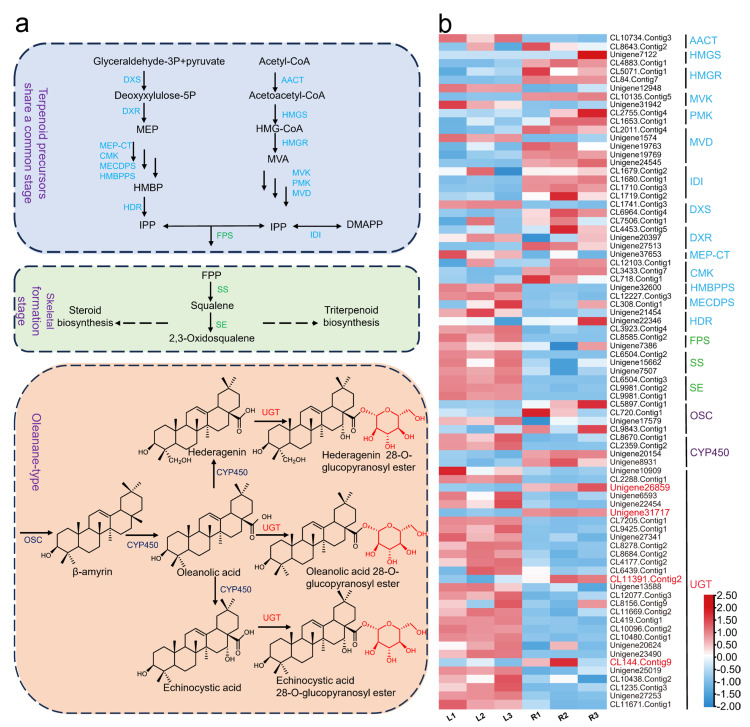
The inferences derived from the biosynthetic pathway of oleanane-type saponins in *H. helix* and the genes participating in this pathway. (**a**) Simplified pathway for the production of oleanane-type saponins is presented. (**b**) Hierarchical clustering is conducted on the expression patterns of 82 unigenes, and the standardized FPKM values are subjected to Z-Score transformation (Appendix A). L1–L3 and R1–R3 represent three repetitions in leaves and roots, respectively. DXS, 1-deoxy-D-xylulose-5-phosphate-synthase; DXR, 1-deoxy-D-xylulose-5-phosphate reductase; MEP-CT, 2-*C*-methyl-D-erythritol-4-phosphate cytidylyltransferase; CMK, 4-(cytidine-5-diphospho)-2-*C*-methyl-D-erythritol kinase; MECDPS, 2-*C*-methyl-D-erythritol-2,4-cyclodiphosphate synthase; HMBPPS, 1-hydroxy-2-methyl-2-E-butenyl-4-diphosphate synthase; HDR, 1-hydroxy-2methy-3-E-butenyl-4-diphosphate reductase; AAC, acetoacetyl-CoA thiolase; HMGS, 3-hydroxy-3-methyl glutaryl coenzyme A synthase; HMGR, 3-hydroxy-3-methyl glutaryl coenzyme A reductase; MVK, mevalonate kinase; PMK, phosphomevalonate kinase; MVD, mevalonate 5-diphosphatcdecarboxylase; IDI, isopentenyl diphosphate isomerase; FPS, farnesyldiphosphate synthase; SS, squalene synthase; SE, squalene epoxidase; OSC, oxidosqualene cyclase; CYP450, cytochrome P450; UGT, UDP-glycosyltransferase.

**Figure 2 ijms-25-04067-f002:**
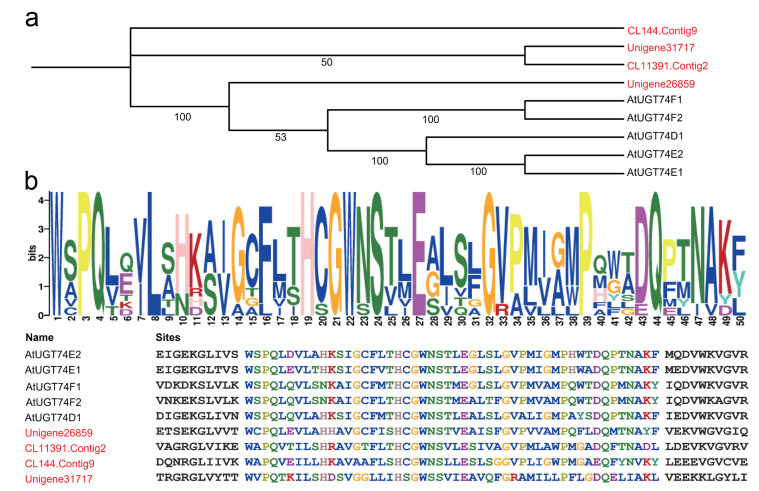
The cluster analysis and PSPG motif alignment of candidate UGTs and *Arabidopsis thaliana AtUGT74* subfamily members. (**a**) The construction of the phylogenetic tree of candidate UGTs and *AtUGT74* subfamily members using MEGA X V1.2.6 software. (**b**) Motif alignment between candidate UGTs and *AtUGT74* with MEME. *H. helix* candidate UGTs are in red font.

**Figure 3 ijms-25-04067-f003:**
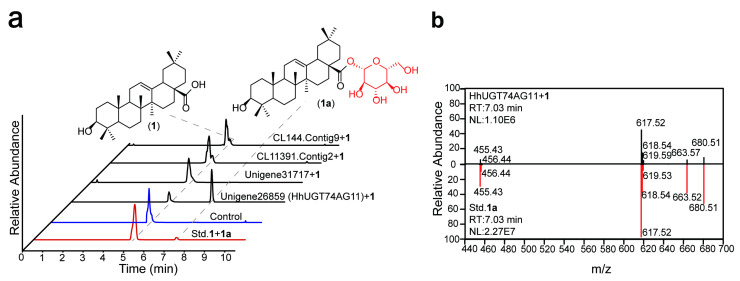
The in vitro enzymatic activity of candidate UGTs detected by UPLC-ESI-MS. (**a**) Extracted ion chromatograms (EICs) for **1** oleanolic acid (*m*/*z* = 454.90–455.90, RT = 4.96 min) and **1a** oleanolic acid 28-*O*-glucopyranosyl ester (*m*/*z* = 616.90–617.90, RT = 7.03 min). (**b**) MS/MS fragmentation of the product peaks by *HhUGT74AG11* + **1** compared with fragmentation results of the Std. **1a** ([(-)-mode], *m*/*z* = 617.52).

**Figure 4 ijms-25-04067-f004:**
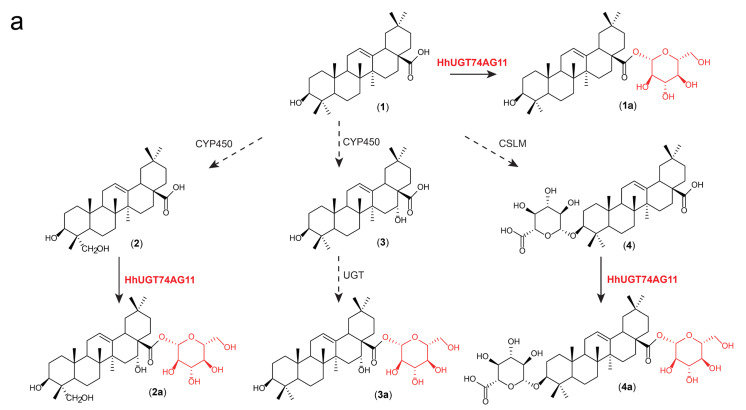
The UPLC-ESI-MS analysis detected the substrate hybridity of *HhUGT74AG11*. (**a**) *HhUGT74AG11* functions in the production of various oleanane-type saponins. (**b**) The identified compounds were as follows: **2** hederagenin (*m*/*z* = 470.90–471.90, RT = 4.69 min) and **2a** hederagenin 28-*O*-glucopyranosyl ester (*m*/*z* = 632.90–633.90, RT = 6.11 min); (**c**) **3** echinocystic acid (*m*/*z* = 470.90–471.90, RT = 4.71 min) and **3a** echinocystic acid 28-*O*-glucopyranosyl ester (*m*/*z* = 632.90–633.90, RT = 6.38 min); (**d**) **4** calenduloside E (*m*/*z* = 630.90–631.90, RT = 4.29 min) and **4a** chikusetsusaponin IVa (*m*/*z* = 792.90–793.90, RT = 4.68 min).

**Figure 5 ijms-25-04067-f005:**
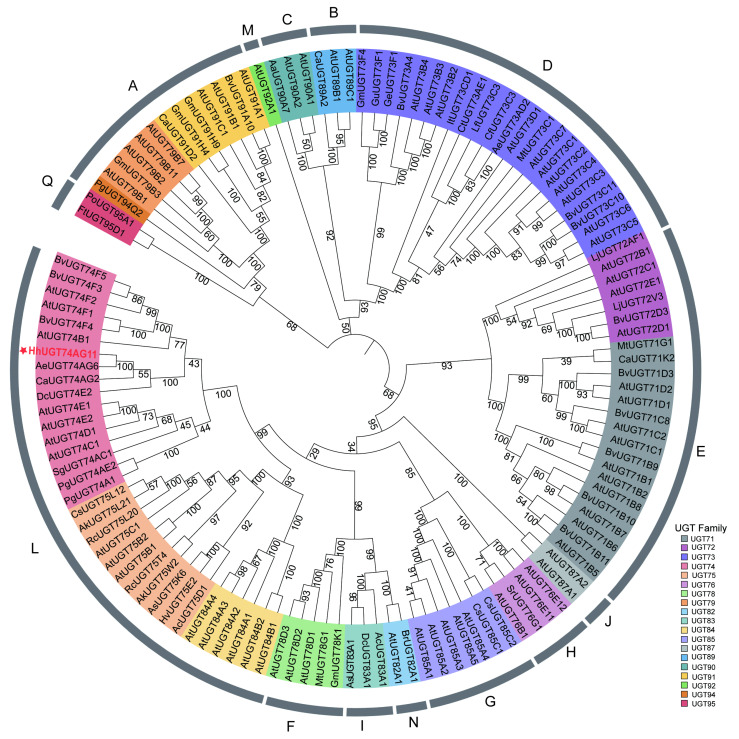
Cluster analysis of *HhUGT74AG11* and UGTs of other plants. Different UGT subfamilies are encoded with diverse colors. The plant species and the accession numbers of the input sequences are provided in Appendix A.

## Data Availability

Data will be made available on request. Sequence data from this article can be found in the GenBank databases under the following accession numbers: *HhUGT74AG11*, OR902350; and the transcriptome data set of *Hedera helix*, SRR5369456.

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
