# Peer review of "Functional Identification of HhUGT74AG11—A Key Glycosyltransferase Involved in Biosynthesis of Oleanane-Type Saponins in Hedera helix"

_ijms, 2024, doi:10.3390/ijms25074067_

Round 1

Reviewer 1 Report

Comments and Suggestions for Authors

Dear the Editor

Yu H et al reported a novel UGT enzyme HhUGT174AG11 by bioinformatic analysis. Subsequently, these authors characterized enzyme products using biochemical technique. Specifically, first an recombinant enzyme was reacted with a variety of substrate in a neutral pH, then, these reaction products were characterized using LC-MS/MS with reversed-phase chromatography. This manuscript appeared well organized, providing a fruitful information for the audience of plant biochemistry.

- In this study, the authors identified a novel gene for Hedera helix UDP-glycosyltransferase. The main question to be addressed was the biochemistry of this enzyme.

- To understand the biochemistry of enzyme HhUGT174AG11 encoded by unigene26859, these authors expressed its recombinant protein and performed LC-MS-based characterization of enzyme products. Prior to this study, nothing was known about Hedera helix UDP-glycosyltransferase, thus this study provided, for the first time, biochemical data for this enzyme.

- The biochemistry of Hedera helix enzyme was first provided by this study.

- Methodology was well-described. An appropriate control was considered.

- Conclusions  appeared reasonable.

- All figures were well organized.

Major concerns:

1)  None.

Minor concerns:

1) Please provide the source of oleanoic acid 28-O-glucopyranosyl ester (LL128-129) in Materials and Method section.

Author Response

Response: Thank you for your valuable comments on our manuscript. We have taken your feedback and made the necessary revisions to improve the quality of our manuscript.

Please provide the source of oleanoic acid 28-O-glucopyranosyl ester (LL128-129) in Materials and Method section.

Response: We feel sorry for our carelessness. In our revised manuscript, the source of oleanolic acid 28-O-glucopyranosyl ester, hederagenin 28-O-glucopyranosyl ester, echinocystic acid 28-O-glucopyranosyl ester and chikusetsusaponin IVa is provided in the Materials and Methods section 4.1. The amended sentence is highlighted in red on lines 278-279 of the manuscript.

Various chemicals, such as echinocystic acid, oleanolic acid, calenduloside E, UDP-Glc, hederagenin, quercetin, luteolin, baicalein, phloretin, genistein, and tectorigenin, oleanolic acid 28-O-glucopyranosyl ester, hederagenin 28-O-glucopyranosyl ester, echinocystic acid 28-O-glucopyranosyl ester and chikusetsusaponin IVa were acquired from Taoshu Biotechnology Company (Shanghai, China) or Yuanye Biotechnology Company (Shanghai, China).

Reviewer 2 Report

Comments and Suggestions for Authors

Title: Functional Identification of HhUGT74AG11 - a Key Glycosyltransferase Involved in Biosynthesis of Oleanane-Type Saponins in Hedera helix

Summary:

This article (ijms-2906835) focuses on identifying a key glycosyltransferase, HhUGT74AG11, involved in the biosynthesis of oleanane-type saponins in Hedera helix, a traditional medicinal plant. Oleanane-type saponins possess significant pharmacological effects, but the enzymes responsible for their biosynthesis were unknown. Through transcriptome analysis and structural domain screening, four candidate UDP-glycosyltransferases (UGTs) were identified, with HhUGT74AG11 being the most promising. In vitro experiments confirmed its ability to convert oleanolic acid into Oleanolic acid 28-O-glucopyranosyl ester and to glycosylate various oleanane-type triterpenoids and flavonoids. Cluster analysis revealed its similarity to functionally identified genes in other plants, supporting its role in substrate generalization. HhUGT74AG11 represents a novel glycosyltransferase crucial for oleanane-type saponin biosynthesis in H. helix, offering insights for the production of valuable triterpenoid saponins.

This paper addresses the specific gap in the field related to understanding the enzymatic processes involved in the biosynthesis of oleanane-type saponins in Hedera hélix, so I think it can be of interest to readers of IJMS. The conclusions drawn in the paper are consistent with the evidence and arguments presented. The main question posed regarding the identification of glycosylation modifying enzymes responsible for oleanane-type saponins biosynthesis is addressed through transcriptome analysis, cluster analysis, and in vitro enzymatic assays, specifically demonstrating the catalytic activity of HhUGT74AG11. The references provided in the article seem appropriate for supporting the research findings and background information. The tables and figures included in the article provide clear visual representations of the experimental results, such as UPLC-MS chromatograms and phylogenetic trees.

Recommendations and specific changes to be made:

·         -Throughout the text: some italics are needed in terms such as “in vitro” or “H. helix”. Examples are in lines 16 and 109.

·         -Throughout the text: Some names are capitalized when they should not be. Example is Oleanolic acid in line 17.

·         -Line 38: A reference is needed in “Oleanane-type saponins are widely applied in clinical practice to treat 37 various respiratory system diseases.”

·         -Line 305, 319 and others: Sometimes "degrees Celsius" is used and other times "ºC" is used. Nomenclature must be unified. Same for other units, specially on Materials and Methods.

·        - I would suggest adding some more details on Materials and Methods 4.2, including details on the sequencing platform used, the assembly software, any quality control measures employed, and how the assembled transcriptome was annotated and analyzed.

·       -  I would suggest separating 4.6 into two different sections, one for in vitro enzyme activity and other for LC-ESI-MS analysis. For in vitro enzyme activity, additional information on the experimental setup, reaction conditions, substrate concentrations, and controls would be valuable for understanding the reliability and reproducibility of the results

Overall, I think that in general it is a good work that only needs to polish some methodological and formatting aspects.

Author Response

Response: Thank you for your valuable comments on our manuscript. We have taken your feedback and made the necessary revisions to improve the quality of our manuscript.

Throughout the text: some italics are needed in terms such as “in vitro” or “H. helix”. Examples are in lines 16 and 109.

Response: Thank you so much for your careful check. We apologize for this mistake. We have revised the plant name H. helix and in vitro to be italicized consistently throughout the text.

Throughout the text: Some names are capitalized when they should not be. Example is Oleanolic acid in line 17.

Response: We deeply apologize for this error. We have modified the article to correctly format the compounds accordingly.

Line 38: A reference is needed in “Oleanane-type saponins are widely applied in clinical practice to treat 37 various respiratory system diseases.”

Response: Thank you for pointing that out. In our revised manuscript, we have added the relevant references on line 38.

Oleanane-type saponins are widely applied in clinical practice to treat various respira-tory system diseases [7].

[7] Lang, C.; Röttger-Lüer, P.; Staiger, C. A valuable option for the treatment of respiratory diseases: review on the clinical evidence of the ivy leaves dry extract EA 575®. Planta medica. 2015, 81(12–13): 968–974.

Line 305, 319 and others: Sometimes "degrees Celsius" is used and other times "ºC" is used. Nomenclature must be unified. Same for other units, specially on Materials and Methods.

Response: Thank you for pointing that out. In the revised manuscript, we have revised the term "degrees Celsius" to "°C" to unify the nomenclature, and the same applies to other units.

I would suggest adding some more details on Materials and Methods 4.2, including details on the sequencing platform used, the assembly software, any quality control measures employed, and how the assembled transcriptome was annotated and analyzed.

Response: Thank you for your suggestion. In our revised manuscript, we have added more details in the Materials and Methods section 4.2.

The sequencing of the library preparations was conducted on an Illumina HiSeq 4000 platform, producing paired-end reads. To derive clean data, sequences with adapter contamination, poly-N stretches, and those of inferior quality were excised from the raw data. High-throughput sequencing and de novo assembly of H. helix transcriptome were based on previous research results [35]. The information was transferred to the SSR database, which is managed by the National Center for Biotechnology Information (NCBI) in the United States. The access number for this upload is SRR5369456. Further examination was conducted on all unigenes identified as associated with the triterpenoid biosynthesis pathway, utilizing databases including Nr, Nt, Swissprot, COG, GO, and KEGG. The FPKM method was used to normalize the ex-pression level of every unigene. Candidate UGT full-length coding sequences were identified by clustering with Arabidopsis thaliana UGT (AtUGT74).

I would suggest separating 4.6 into two different sections, one for in vitro enzyme activity and other for LC-ESI-MS analysis. For in vitro enzyme activity, additional information on the experimental setup, reaction conditions, substrate concentrations, and controls would be valuable for understanding the reliability and reproducibility of the results.

Response: Thank you for your suggestion. In the revised manuscript, we have split section 4.6 into two distinct parts: section 4.6 is dedicated to in vitro enzyme activity determination, and section 4.7 is used for UPLC-ESI-MS analysis.

4.6. In vitro enzyme activity determination

The assay to measure the enzyme activity of the recombinant HhUGT74AG11 was conducted in a reaction mixture of 200 μL, which consisted of 50 mM Tris-HCl (pH 7.0), 0.1 mM MgCl2, 0.1 mM substrate for the receptor, 1 mM UDP-Glc, and 10 μg of the pro-tein that had been purified. The incubation was carried out at 30°C for 12 hours. The reaction was stopped by adding 200 μL of 80% methanol. The negative control reaction used boiled recombinant proteins (100°C and 15 min). After centrifugation at a speed of 12000 revolutions per min for a duration of 2 min, the reaction mixture was then passed through a membrane filter with a pore size of 0.22 μm. Then, machine detection was carried out on it with HPLC or UPLC-ESI-MS.

4.7. UPLC-ESI-MS analysis

The examination of flavonoid reaction products was carried out utilizing HPLC (Thermo Fisher, USA) equipped with a WondaCract ODS-2 C18 column (4.6 * 250 mm; particle size 5.0 μm granularity; Shimadzu, Japan). Solvent A, which was acetonitrile, and solvent B, which was a mixture of 0.1% formic acid in water, made up the mobile phase. The elution process is carried out in the following manner: solvent A is used at a concentration of 20% for 0 min, followed by 50% for 10 min, and finally 100% for 18 min. The rate of flow is adjusted to 1.0 milliliters per min, accompanied by an injection volume of 5 microliters, while maintaining a temperature of 35°C. Spectral detection was performed at 269 nm using a diode array detector (DAD).

Reaction products were analysed by UPLC-ESI-MS after storage of the Shim-pack GIST-HP C18 column at 35°C. The separation of Acetonitrile (solvent A) and 0.1% for-mic acid in water (solvent B) occurred with a flow rate of 0.2 mL/min, while maintain-ing an injection volume of 1 μL. The experimental conditions were as follows: 0 minutes with 5% solvent A, 0.5 min with 5% solvent A, 4.5 min with 95% sol-vent A, 7.5 min with 95% solvent A, 7.6 min with 5% solvent A, and 10.5 min with 5% solvent A. All mass spectrometry data were acquired using the TSQ quantification ™ Triple Quadrupole Mass Spectrometer fitted with an ESI probe inlets (Thermo Fisher, USA). The parameters for ESI source are as follows: triterpenoids are examined using negative ionization mode; flavonoids are examined using positive ion-ization mode; the spray voltage is set at 3.5 kV; collision energy is set at 35 V; the scan-ning range is from 260 to 850 m/z. The flow of the auxiliary gas (also known as sheath gas) was adjusted to 40 arb, while the flow of the auxiliary/scavenging gas (specifically nitrogen) was set to 10 arb. The temperature of the auxiliary gas heater was kept con-stant at 300°C, and the capillary temperature was set to 320°C.

Overall, I think that in general it is a good work that only needs to polish some methodological and formatting aspects.

Response: Thank you for your valuable feedback on our work. We appreciate your positive assessment and acknowledge the areas where improvements can be made in terms of methodology and formatting. We will address these suggestions to enhance the overall quality of the manuscript. If you have any specific recommendations or further guidance on these aspects, we would be grateful for your insights to refine our work. Thank you once again for your time and constructive comments.

Reviewer 3 Report

Comments and Suggestions for Authors

The text bellow contains comments on manuscript entitled “Functional Identification of HhUGT74AG11-a Key Glycosyl-transferase Involved in Biosynthesis of Oleanane-Type Saponins in Hedera helix”.

The manuscript is focused on the identification of glycosylation modifying enzymes responsible for catalyzing oleanane-type saponins biosynthesis in H. helix by using transcriptome, cluster analysis and PSPG structural domain.

The manuscript  is well written with a logically designed experimental scheme. The obtained results fully explain the aim of the study.

Here, I am listing several suggestions for minor correction that the authors might take into consideration:

Page 1, Lines 34-35: The name of the metabolites should be in small letters, unless they stand at the beginning of the sentence.

Page 2, Lines 66-80: Please present in a more clear way the aim of your study, what is the novelty in it and your expectations.

Page 2, Line 84: The name of the plant H. helix must be in italic fonts in the whole text.

Page 9, Line 250: Written: degrees Celsius. Please write it with the relevant symbols.

Page 10, Lines 290-291: Please specify which sequence of the primers is forward and which one is reverse.

Comments on the Quality of English Language

Minor editing of English language required

Author Response

Response: Thank you for your valuable comments on our manuscript. We have taken your feedback and made the necessary revisions to improve the quality of our manuscript.

Page 1, Lines 34-35: The name of the metabolites should be in small letters, unless they stand at the beginning of the sentence.

Response: We apologize for this mistake. We have revised the article to correctly format the metabolites accordingly.

Page 2, Lines 66-80: Please present in a more clear way the aim of your study, what is the novelty in it and your expectations.

Response: Thank you for your advice. In the revised manuscript, we strive to simplify the description at the end of the introduction as much as possible. The revised sentences have been highlighted in red on lines 84-97 of the manuscript.

In this study, four candidate UGT genes potentially involved in the biosynthesis of oleananes-type triterpenoids in H. helix were preliminarily screened through tran-scriptome, cluster analysis, and PSPG domain. Using the Escherichia coli expression sys-tem and protein purification methods, the in vitro enzymatic activity of the candidate UGT genes was tested through precursor feeding assays. The results showed that only Unigene26859 (HhUGT74AG11) was active towards oleanolic acid. In addition, consid-ering the diversity of the structures of oleananes-type triterpenoids, we also tested a variety of triterpenoid compounds and flavonoids, and HhUGT74AG11 exhibited broad substrate promiscuity. Combining cluster analysis, we further discussed the reasons for the interesting phenomenon of the functional evolution of HhUGT74AG11 and the structural diversification of secondary metabolites. In summary, this study cloned and functionally identified the glycosyltransferase HhUGT74AG11 from H. helix, revealing its outstanding contribution to the biosynthesis, laying the foundation for the development of drugs with diverse structures of oleananes-type triterpenoids.

Page 2, Line 84: The name of the plant H. helix must be in italic fonts in the whole text.

Response: Thank you so much for your careful check. We apologize for this mistake. We have revised the plant name H. helix to be italicized consistently throughout the text.

Page 9, Line 250: Written: degrees Celsius. Please write it with the relevant symbols.

Response: Thank you for pointing that out. In the revised manuscript, we have revised the term "degrees Celsius" to "°C" to unify the nomenclature, and the same applies to other units.

The samples were rapidly cryopreserved using liquid nitrogen and then stored at -80°C for later analysis. There were three biological replicates for each sample.

Page 10, Lines 290-291: Please specify which sequence of the primers is forward and which one is reverse.

Response: We feel sorry for our carelessness. In our revised manuscript, the forward and reverse primers are provided in the supplementary Table S1.

Table S1. Primers used in this study.

Primer

Sequences (5’-3’)

Unigene26859 (HhUGT74AG11)- Forward primer

gatctggttccgcgtggatccATGGAGAATGAGAAAACTTATAAAGCTC

Unigene26859 (HhUGT74AG11)- Reverse primer

ctcgagtcgacccgggaattcTTAGAGTGCCAAAATCCGAGAAA

Unigene31717-

Forward primer

gatctggttccgcgtggatccATGGCCGCGAATGACAAA

Unigene31717-

Reverse primer

ctcgagtcgacccgggaattcTCATTTTTTGAGGATTTTATGATTTTC

CL11391.Contig2- Forward primer

gatctggttccgcgtggatccATGGCGGTGGCCGGCGCC

CL11391.Contig2- Reverse primer

ctcgagtcgacccgggaattcTTAGTATTCAGATAAATGTCTAACCAACTCA

CL144.Contig9-

Forward primer

gatctggttccgcgtggatccATGGAAGAGAGAAAAGGAAAGACCA

CL144.Contig9-

Reverse primer

ctcgagtcgacccgggaattcCTATTCATTCTCCCCTCTTGGCT

Minor editing of English language required

Response: Thank you for your feedback. We have made the necessary minor edits to improve the English language in the manuscript. Your input is greatly appreciated, and we are committed to enhancing the clarity and quality of the writing. If you have any specific suggestions or areas of concern regarding the language, please feel free to let us know. Thank you for your valuable input, and we will ensure that the revised version meets the required standards.

Round 2

Reviewer 1 Report

Comments and Suggestions for Authors

Dear the Editor

The revised manuscript properly addressed the raised concerns by this Reviewer.